# Platyphylloside Isolated from *Betula platyphylla* is Antiproliferative and Induces Apoptosis in Colon Cancer and Leukemic Cells

**DOI:** 10.3390/molecules24162960

**Published:** 2019-08-15

**Authors:** Joo-Eun Lee, Nguyen Thi Thanh Thuy, Jina Lee, Namki Cho, Hee Min Yoo

**Affiliations:** 1Stem Cell Research Center, Korea Research Institute of Bioscience and Biotechnology (KRIBB), Daejeon 34141, Korea; 2College of Pharmacy, Chonnam National University, Gwangju 61186, Korea; 3Center for Bioanalysis, Korea Research Institute of Standards and Science (KRISS), Daejeon 34113, Korea

**Keywords:** platyphylloside, *Betula platyphylla*, apoptosis, reactive oxygen species (ROS), Jurkat cells

## Abstract

*Betula platyphylla* bark has been evaluated for the treatment of dermatitis, inflammatory conditions, and cancer. Diarylheptanoids are the major constituents of the *B. platyphylla* bark and possess various pharmacological effects. Our previous study confirmed the selective antiproliferative effect of platyphylloside (BPP) isolated from *B. platyphylla* on colon cancer and leukemic cells using 60 different cancer cell lines from thr National Cancer Institution (NCI). In line with previous reports, this study focuses on the apoptotic pathway of BPP, a phenolic glycoside composed of two aromatic rings joined by a seven-carbon chain. Cytotoxicity assays in solid tumor and blood cancer cell models demonstrated that BPP possesses potent antiproliferative activity. The level of apoptosis increased with BPP treatment, causing cell cycle arrest at the G1 phase along with the downregulation of IκBα phosphorylation and BCL-2, as well as upregulation of cleaved caspase 3 and BAX proteins. In addition, BPP displayed potent mitochondrial depolarization effects in Jurkat cells. The combined findings revealed that the cytotoxic effects of BPP were mediated by intracellular signaling, possibly through a mechanism involving the upregulation of mitochondrial reactive oxygen species (ROS). Thus, BPP could be a potential multitarget therapeutic agent in leukemia and colon cancer.

## 1. Introduction

Cancer is the second leading cause of death in the United States, and colorectal cancer (CRC) is one of the most common cancers in both men and women worldwide [1]. In Korea, cancer headed the list of the top 10 leading causes of death, and colorectal cancer (CRC) is the third most common cancer in men and women, with more than one million cases diagnosed each year worldwide [2]. Chemotherapeutic regimens, including 5-fluorouracil (5-FU), oxaliplatin, and irinotecan, are reportedly efficacious in colon cancer [3]. Additional targeted agents, including anti-EGFR or anti-VEGF antibodies, are considered to be effective in metastatic CRC cases (mCRC) [3]. Recently, research efforts in CRC drug discovery have focused on compounds derived from natural products due to the limited associated side effects [4]. Leukemia is a type of blood cancer that usually begins in the bone marrow and results in high abnormal blood cell counts. It is categorized according to how quickly the disease develops (acute or chronic) and the blood cell type affected (lymphocytes or myelocytes). The four main leukemia types include acute lymphocytic leukemia or acute lymphoblastic leukemia (ALL), chronic lymphocytic leukemia (CLL), acute myelocytic leukemia (AML), and chronic myelocytic leukemia or chronic myelogenous leukemia (CML). Leukemia treatments include chemotherapy, radiation and targeted therapies, and stem cell transplants. However, the conventional therapies are limited by the associated high costs and anticancer drug toxicity, challenging researchers to develop biocompatible and cost-effective new drugs [5,6,7]. Recent findings have demonstrated the considerable potential of alternative interventions, such as immunotherapy and natural products (NPs), in the treatment of leukemia [5,6,7]. NPs are capable of targeting multiple cancers and may provide a more robust treatment of cancer by limiting treatment-acquired resistance, increasing the efficacy of individual components in a cancer therapy cocktail, and reducing the associated side-effects to achieve a positive treatment response [5,6,7].

A platyphylloside denoted as BPP (*Betula platyphylla* platyphylloside) is a well-known diarylheptanoid obtained from the bark of *Betula platyphylla* (birch tree), which is widely distributed in Korea, Japan, China, Sahalin island, and Siberia [8]. The healing properties of *B. platyphylla* bark have long been known in traditional medicine located in different parts of the world [8,9]. Its most widespread uses have been in inflammatory diseases, including dermatitis, bronchitis, and cancer treatment [8,9]. *B. platyphylla* diarylheptanoids are phenolic compounds composed of two aromatic rings joined by a seven-carbon chain, containing various phenylpropanoid pathway-derived substituents. These compounds exhibit a wide range of biological activities, including antioxidant, anti-inflammatory, and anticancer activities [10,11,12,13,14,15,16,17,18,19]. Two major *B. platyphylla* diarylheptanoids, previously isolated with an efficient high speed counter current chromatography (HSCCC) method, showed antiproliferative potential in 60 National Cancer Institute (NCI) cancer cell lines [20]. Our previous study reported that platyphylloside—one of the two major *B. platyphylla* bark diarylheptanoids [19]—showed potent cytotoxic activities against several cancer cell lines, with selectively towards colon cancer and leukemic cells [20]. However, BPP’s mechanism of action in colon and leukemia cell lines has not been established to date. Herein, BPP’s antiproliferative and cell cycle effects on colorectal and blood cancer cells were determined. Furthermore, BPP-induced apoptosis was studied to understand its antiproliferative mechanism of action.

## 2. Results and Discussion

RKO, the human colorectal cancer cell line, was used to examine the antitumor effect of BPP [21]. Furthermore, human leukemic Jurkat T-cells were used to assess the antiproliferative and proapoptotic activity of BPP [22]. In this study, we employed solid tumor and blood cancer cell lines to evaluate the potential of BPP as a multitarget therapeutic agent, in both colon cancer and leukemia. To evaluate the mechanism of BPP in vitro, a large amount of BPP (Figure 1A) was isolated from the CH_2_Cl_2_ fraction of the *B. platyphylla* bark extract using high-speed counter-current chromatography, as previously described [20]. In terms of the structure–activity relationships, in vitro results from previous studies suggest that the structure of BPP, which consist of two aromatic rings joined by a seven carbon chain along with a glucosyl group at C-5 and a carbonyl group at C-3, positively correlated with anticancer activities [20,23].

### 2.1. BPP-Induced Antiproliferative Effects in Colon Cancer Cells

To determine the antiproliferative potential of BPP, it was screened against multiple colon cancer cell lines (Appendix A), using a cell proliferation assay (CellTiter Glo). BPP showed a pronounced antiproliferative activity against RKO. The antiproliferative effect increased with the dose (20 μM) and treatment duration of 48 h (Figure 1B). After BPP (20 μM) treatment, cell viability significantly reduced. Microscopic analysis also showed that BPP treatment induced morphological changes compared to the dimethyl sulfoxide (DMSO) treatment (Figure 1C). Considering that BPP showed a slightly stronger cytotoxicity against RKO than the other multiple colon cancer cell lines (Appendix A), our subsequent studies focused on the mechanism of action of BPP in RKO.

### 2.2. BPP-Induced Apoptosis in RKO Cells

Moreover, BPP-induced apoptosis was determined using the RKO cell line. RKO cells were double-stained with annexin V /7-AAD dyes to determine the early and late apoptotic and viable cell percentages. The percentages of early (Annexin^+^/7-AAD^–^) and late apoptosis (Annexin^+^/7-AAD^+^) cells increased and the live cells decreased, 24 h post BPP treatment, confirming the induction of apoptosis (Figure 2A,B). BPP-treated RKO cells showed a dramatic increase in early (43.3%) apoptotic cell numbers within 48 h of treatment (Figure 2A). Apoptosis-related protein levels in RKO cells in response to BPP (20 μM) treatment were also analyzed using western blots (WB). We found that BPP downregulated the antiapoptotic protein BCL2, upregulated BAX, and cleaved caspase 3 expression (Figure 2C). To define more molecular mechanisms underlying BPP-induced cell apoptosis, we investigated NF-κB signaling by evaluating the phosphorylation of IκBα protein. NF-κB is a critical transcription factor that regulates many genes associated with tumorigenesis [24]. Our data indicate that the phosphorylation of the inhibitory protein, IκBα, was reduced by BPP (Figure 2D).

### 2.3. BPP-Induced Cell Cycle Arrest in RKO Cells

The inhibition of cell viability by BPP, due to effects on cell cycle progression, was investigated in RKO cells (Figure 3A,B). RKO incubation with BPP resulted in an increased sub-G1 and decreased S phase cell percentages. Similar results have been reported in colon cancer cells treated with curcumin, a diarylheptanoid isolated from Curcuma longa [25]. Previous in vitro studies suggest that, compared with diarylheptanoids, the coexistence of a glucosyl group at C-5 and a carbonyl group at C-3 in these diarylheptanoids is vital, and confers significant activities and selectivity towards cancer cells [23]. Our results also prove that BPP inhibits colon cancer cell growth by inducing cell cycle arrest in RKO cells.

### 2.4. BPP-Induced Antiproliferative Effects in Leukemic Cells

The antiproliferative activity of BPP has been reported in solid tumors such as colorectal cancer (CRC), but the effect is largely unexplored in blood cancers such as leukemia. This study, to our knowledge, is the first to evaluate the anticancer effect of BPP in blood cancer using Jurkat leukemia cell lines. To further assess the growth inhibition properties of BPP, a dose-dependent MTS screening assay was performed in multiple blood cancer cell lines (Figure 4A, Appendix A). Dose-dependent growth inhibition was observed in each cell line, with the effect being notably stronger in Jurkat cells. Compared to dimethyl sulfoxide (DMSO) treatment, BPP treatment induced morphological changes. BPP treatment significantly reduced viability of Jurkat leukemia cells, but DMSO-pretreatment markedly reversed these cytotoxic effects (Figure 4B).

### 2.5. BPP-Induced Apoptosis in Jurkat Cells

To examine BPP-induced cell death in relation to apoptosis, we used the annexin V/7-AAD double staining-based fluorescence-activated cell sorting analysis. BPP-treated Jurkat cells showed a dramatic increase in both early (12.3%) and late (72.5%) apoptotic cell numbers within 48 h of treatment (Figure 5A). However, this apoptosis was reversed in the DMSO control (Figure 5B). As the apoptotic cell population dramatically increased, we investigated the relative apoptosis-associated protein expression levels. Western blots revealed that BPP downregulated the expression of the antiapoptotic protein BCL-2, a survival factor in apoptosis regulation and upregulated cleavage of poly (ADP-ribose) polymerase (PARP), which promotes apoptosis by preventing DNA repair, and cleaved caspase 3 (Figure 5C). In addition, BPP induced cell apoptosis in Jurkat cells through the downregulation of p-IκBα, which is related to NF-κB signaling, a critical transcription factor that regulates many genes associated with tumorigenesis (Figure 5D).

### 2.6. BPP-Induced Cell Cycle Arrest in Jurkat Cells

We also found that BPP markedly increased the sub-G1 cell population. BPP (20 μM) treatment for 48 h caused accumulation of cells in the sub-G1 phase, from 70.7% in the untreated control cells to 78.0% following BPP treatment (Figure 6A,B). A similar result was reported for Jurkat cells treated with curcumin [26]. Overall, these results of BPP-induced accumulation of apoptotic leukemia cell populations reveal that BPP may be responsible for the inhibition of cell growth via induction of cell cycle arrest at the sub-G1 phase.

### 2.7. BPP Effects on Mitochondrial Membrane Potential in Apoptotic Cells

Next, BPP effects on the mitochondrial membrane potential (ΔΨm) were evaluated by probing tetramethylrhodamine methyl ester (TMRM^+^) fluorescence intensity using flow cytometry. TMRM^+^ quantifies changes in the live cell mitochondrial membrane potential. Figure 7A shows representative data from DMSO-treated negative controls and BPP-treated Jurkat cells. BPP-treated (20 μM) cells showed significant membrane potential depolarization after 48 h (Figure 7B). Thus, BPP showed mitochondrial depolarization properties in Jurkat leukemic cells. These results suggest that *B. platyphylla* BPP could be a potential chemotherapeutic agent for blood cancer.

### 2.8. BPP-Induced ROS in Jurkat Cells

Mitochondria are major producers of reactive oxygen species (ROS) in cancer cells [27,28]. An elevated redox imbalance, relative to normal cells, is a common feature of cancer progression [29]. Association of ROS production with apoptosis and cell cycle arrest caused by anticancer agents has been demonstrated [28,29]. To examine how BPP causes cell death, ROS levels in each well were evaluated using the 2′,7′-dichlorodihydrofluorescein diacetate (H2DCFDA) assay.

Using confocal microscopy, we observed that BPP increased intracellular ROS levels in Jurkat cells after 48 h (Figure 8A). ROS generation was quantified by flow cytometry. Similar to the results observed by fluorescence microscopy, ROS generation in Jurkat cells was increased by BPP treatment (Figure 8B,C). Collectively, our results suggest that ROS generation by BPP is a potential mechanism underlying ROS-dependent apoptosis in Jurkat leukemic cells.

### 2.9. Live/Dead Assay in Jurkat Cells

The live/dead viability/cytotoxicity assay is most commonly used to validate and quantify live and dead cell populations, which involves labeling them with calcein (green) and ethidium homodimer-1 (red) fluorophores [30]. The live/dead assay was imaged by fluorescence microscopy (Figure 9A). As is typical of the live/dead stain, the BPP-exposed population demonstrated red fluorescence (white arrow) and weak green fluorescence. BPP-treated Jurkat cells showed a dramatic increase in red fluorescence-labeled dead cells within 48 h of treatment. However, this observation was reversed in the DMSO control (Figure 9B). These results corroborate the above results in which BPP significantly elevated the proportion of apoptotic cells in this study.

## 3. Materials and Methods

### 3.1. Plant Material

*Betula platyphylla* bark was obtained from the afforested land of SK E & C (Seoul, Korea), and a voucher specimen (SNU-797) was deposited in the Herbarium of the Medicinal Plant Garden, College of Pharmacy, Seoul National University, Korea. High yield platyphylloside (5.0%) was isolated from the *Betula platyphylla* CH_2_Cl_2_ fraction using high-speed counter-current chromatography, as previously descried [20].

### 3.2. Cell Culture

RKO, WiDr, HT-29, SW48, SW480, SW620, LoVo, HCT-116, DLD-1, LS174T, Jurkat, Daudi, and HEL 92.1.7 cells were obtained from the American Type Culture Collection (ATCC, Manassas, VA, USA). OCI-LY3 was obtained from The European Molecular biology laboratory (DSMZ, Braunschweig, Germany). The culture medium for each cell line was prepared according to the ATCC protocol. Cell lines were cultured in a humidified atmosphere of 5% CO_2_ at 37 °C and subcultured at a ratio of 1:5, when the cell density reached 80–90%, every 3 or 4 days.

### 3.3. Evaluating BPP-Induced Antiproliferation of Colon Cancer and Leukemic Cells

The antiproliferative effect of BPP on colon and blood cancer cells was measured using the CellTiter 96 AQueous One Solution Cell Proliferation Assay Kit (Promega, Madison, WI, USA). The cells were cultured at a density of 5 × 10^5^ cells/well in RPMI 1640 (100 μL) for 24 h. The cells were then treated with different BPP concentrations (0 μM, 1 μM, 3 μM, 5 μM, 6 μM, 10 μM, 20 μM, 50 μM, and 100 μM). DMSO-treated cells served as the control. To estimate cell viability (%) 48 h post BPP treatment, the cells were incubated with 20 μL of CellTiter 96 AQueous One Solution Cell Proliferation Assay reagent for 2 h at 37 °C. The optical density (OD) at 490 nm was measured using a microplate reader (Synergy HTX, BIO-TEK Instruments, Inc., Winooski, VT, USA).

### 3.4. Microscopy

BPP-treated RKO and Jurkat cells were examined under a phase contrast microscope (Olympus, Tokyo, Japan) to detect morphological changes. Photomicrographs of the leukemic cells were taken (at 400×) 24 and 48 h post BPP treatment, and the cells were analyzed for changes in shape, size, and number.

### 3.5. Cytotoxicity Assay

To explain the BPP-induced cytotoxicity mechanism in RKO and Jurkat cells, we investigated changes in cell cycle progression and apoptosis onset using flow cytometry.

### 3.6. Determination of BPP-Induced Apoptosis

BPP-treated RKO and Jurkat cells were visualized by flow cytometry using the PE Annexin V Apoptosis Detection Kit with 7-AAD (BioLegend, San Diego, CA, USA). Annexin V binds cell phosphatidylserine (PS) as PS flips from the inner to the outer leaflet of the plasma membrane during induction of early apoptosis induction. In contrast, PI stains DNA when the cell membrane is disrupted during late apoptosis. To distinguish PE from PI fluorescence and discriminate between early (Annexin V^+^) and late apoptotic cell (7-AAD^+^) populations, FL-2 and FL-3 channels were selected to measure the PE and 7-AAD fluorescence, respectively. BPP-treated and untreated (control) RKO and Jurkat cells were seeded in 6-well plates and incubated at 37 °C for 48 h. Cells were harvested 48 h post-treatment, washed with phosphate-buffered saline (PBS), and resuspended in binding buffer (1×). Cells were stained with annexin V/7-AAD, per the manufacturer’s instructions. After staining, the cells were discriminated as early (Annexin V^+^/7-AAD^–^) and late (Annexin V^+^/7-AAD^+^) apoptotic cells using a flow cytometer (BD FACSVerse, BD Biosciences, San Jose, CA, USA). Cell percentages were analyzed using the FlowJo software (Becton, Dickinson and Company, Franklin Lakes, NJ, USA).

### 3.7. Western Blot

Cell lysates were prepared with lysis buffer containing 25 mM of Tris-HCl (pH 7.6), 150 mM of NaCl, 1% NP-40, 1% sodium deoxycholate, and 0.1% sodium dodecyl sulfate, supplemented with a protease inhibitor cocktail (complete, Roche Molecular Biochemicals, Basel, Swtizerland). Lysates were placed on ice for 5 min, and then centrifuged at 13,000 rpm at 4 °C for 10 min. Total protein concentrations were measured with Pierce BCA protein assay (ThermoFisher Scientific, Waltham, MA, USA). Total protein lysates (40 μg per lane) were separated by sodium dodecyl sulfate-polyacrylamide gel electrophoresis (SDS-PAGE) on 4–12% Bis Tris-NuPage polyacrylamide gels using 1× MES as running buffer (ThermoFisher Scientific, Waltham, MA, USA), and transferred to PVDF membranes (0.45 μM pore size,). After blocking nonspecific binding with 5% nonfat dry milk, membranes were probed with the following monoclonal or polyclonal primary antibodies, according to the manufacturer’s recommendations: anti-cleaved PARP-1, anti-BCL-2, anti-BAX, I-kappa-B-alpha (IκBα), phosphorylated I-kappa-B-alpha (p-IκBα), anti-actin (Santa Cruz Biotechnology, Inc., Dallas, TX, USA), and anti-cleaved caspase 3 (Cell signaling Techonology, Danvers, MA, USA). For detection, membranes were incubated with HRP-conjugated secondary antibodies (Jackson Laboratory, Bar Harbor, ME, USA), at 1:5000 dilution each, scanned with an ImageQuant LAS 4000 mini (Fujifilm, Tokyo, Japan), and analyzed with an image analysis program (Multi Gauge Ver. 3.0, Fujifilm, Tokyo, Japan).

### 3.8. Tetramethylrhodamine Methyl Ester Perchlorate (TMRM) Assay

BPP-treated Jurkat cells were harvested 48 h post-incubation. The cells were washed with PBS and incubated with 100 nM cell-permeable fluorescent indicator TMRM (ThermoFisher Scientific, Waltham, MA, USA), prepared by diluting 10 mM stock in PBS for 30 min at 37 °C. For mitochondrial membrane potential analysis, the cells were washed and resuspended in PBS, measured with a flow cytometer (BD FACSVerse, BD Biosciences, San Jose, CA, USA), and analyzed using the FlowJo v10 software (Becton, Dickinson and Company, Franklin Lakes, NJ, USA)

### 3.9. Evaluating BPP-Induced Changes in Cell Cycle Progression

Univariate analysis of the cellular DNA content of BPP-treated cells enables the detection of cells in different cycle phases (G0/G1, S, or G2/M) and apoptotic cells with fractional DNA content. The BPP-treated RKO and Jurkat cells were seeded into 6-well plates and incubated for 48 h. The cells were harvested, washed with PBS, and fixed with ice-cold 70% ethanol for 3 h at 4 °C. For cell cycle analysis, the cells were resuspended in PBS containing RNase A (5 μg/mL) and PI (50 μg/mL). The cells were evaluated using a flow cytometer (BD FACSVerse, BD Biosciences, San Jose, CA, USA) and cell percentages G0/G1, S, and G2/M phases were analyzed using the FlowJo software (Becton, Dickinson and Company, Franklin Lakes, NJ, USA).

### 3.10. Measuring Reactive Oxygen Species (ROS)

Intracellular ROS levels were detected using 2′,7′-dichlorodihydrofluorescein diacetate acetyl ester (H2DCFDA; Thermo Fisher Scientific, Waltham, MA, USA). Cells continuously perfused with PBS buffer (37 °C) were imaged using an Inverted fluorescent microscope (Olympus). For this analysis, cells were incubated with DCFDA (1 μM) for 30 min at room temperature. Cells were also washed with cold PBS and resuspended in 0.5 mL of PBS supplemented with 1% fetal bovine serum. Intracellular fluorescence accumulation was analyzed using a flow cytometer (BD FACSVerse, BD Biosciences, San Jose, CA, USA).

### 3.11. Live/Dead Fluorescence Microscopy Assay

Jurkat cell morphology was investigated by the simultaneous fluorescent labeling of both living and dead cells using the LIVE/DEAD kit (Thermo Fisher Scientific, Waltham, MA, USA). They were stained for 20 min in the dark using a calcein AM and ethidium homodimer (EthD-1), as per the manufacturer’s instructions. Images were captured using a fluorescence microscope (Olympus, Tokyo, Japan).

### 3.12. Statistical Analyses

Statistical analyses were performed using GraphPad Prism 5 (GraphPad Software, Inc., San Diego, CA, USA), and the values are presented as the mean ± S.D. The results were further analyzed by one-way ANOVA and t-test, *p*-values < 0.05 were considered statistically significant.

## 4. Conclusions

Platyphyolloside (5-hydroxy-1,7-bis-(4-hydroxy-phenyl)-3-heptan-one-3-O-β-dglucopyranoside) is an arylheptanoid glucoside. Its mechanism of action in colon and leukemia cell lines has never been previously reported. In this study, we show that platyphylloside inhibits cell growth by inducing cell cycle arrest and apoptosis in both colorectal and leukemia cell lines, targeting more than one cancer-related mechanism to exert its antiproliferative activity. We investigated the cell cycle effect, and our results suggest that cell cycle arrest plays an important role in the inhibitory effect of BPP in RKO colon and Jurkat leukemia cells by preventing NF-κB signaling. Furthermore, platyphylloside induced mitochondrial depolarization and increased apoptosis in Jurkat cells. Furthermore, platyphylloside upregulated mitochondrial ROS to potentiate the cytotoxic effects via intracellular signaling. Although further studies are required to evaluate its in vivo anticancer activity, we were able to demonstrate that platyphylloside can be a potent multitarget anticancer chemotherapeutic agent by inducing cell cycle arrest and apoptosis via a ROS-dependent pathway.

## Figures and Tables

**Figure 1 molecules-24-02960-f001:**
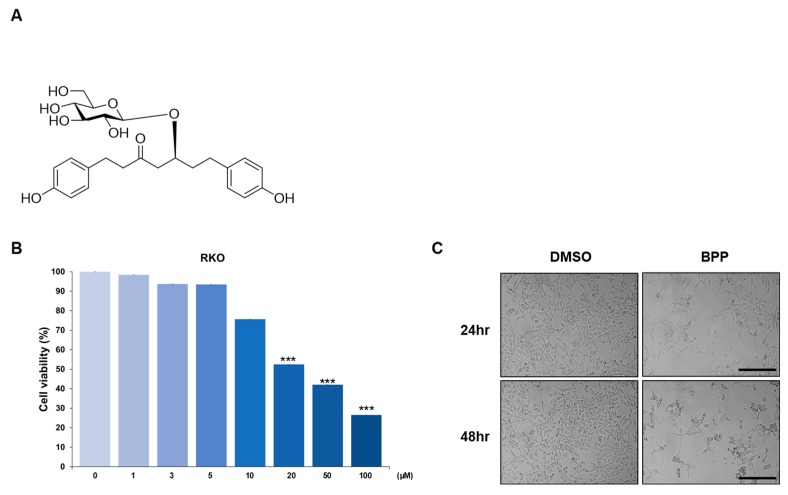
(**A**) Chemical structure of BPP (*Betula platyphylla* platyphylloside). (**B**) BPP showed an antiproliferative effect on RKO cells. Cell viability was determined using the MTS (3-(4,5-dimethylthiazol-2-yl)-5-(3-carboxymethoxyphenyl)-2-(4-sulfophenyl)-2H-tetrazolium, inner salt) assay 48 h post treatment with the compound. (**C**) The BPP (20 μM)-treated RKO (human colorectal cancer cell line) cells showed morphological changes characteristic of apoptosis. Phase contrast microscopy images (400× magnification) showing morphological changes 24 and 48 h post compound treatment.

**Figure 2 molecules-24-02960-f002:**
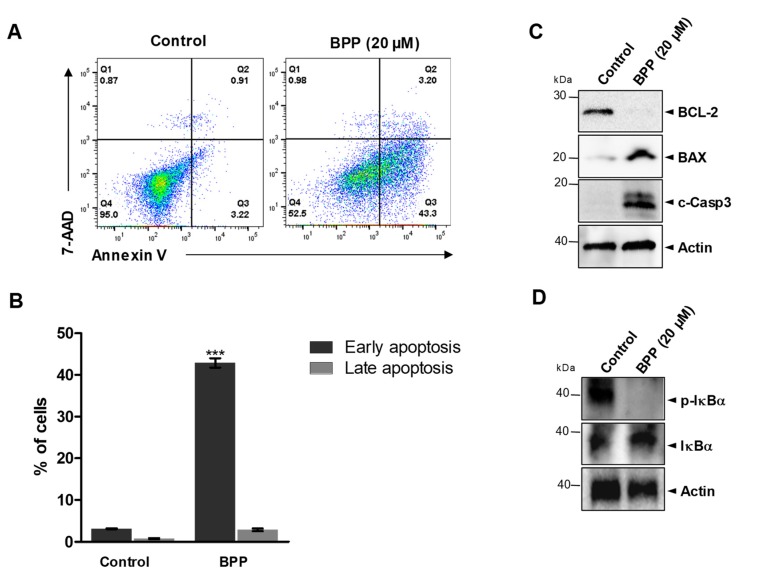
BPP-induced apoptosis in RKO cells. (**A**) RKO cells were compound-untreated or -treated, and apoptosis was evaluated 48 h later using annexin V/7AAD double staining. Cell death was assessed by flow cytometry. (**B**) Quantification of the early and late apoptotic cell rates. (**C**) Whole cell lysates from BPP-treated cells were immunoblotted with antibodies specific for BCL-2, BAX, and caspase-3 proteins. (**D**) Whole cell lysates were used to determine the expression levels of I-kappa-B-alpha (IκBα) and p-IκBα after treating the cells with BPP (20 μM).

**Figure 3 molecules-24-02960-f003:**
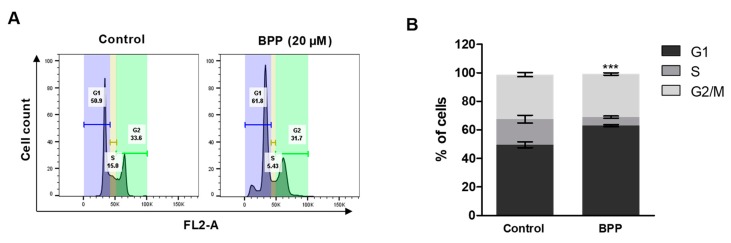
BPP-induced changes in RKO cell cycle progression. (**A**) Cell cycle analysis of compound-treated cells. Cells were stained with PI (propidium iodide) for 30 min and analyzed by flow cytometry. (**B**) Percentage G1 or S, G2/M phase cells from triplicate experiments were determined.

**Figure 4 molecules-24-02960-f004:**
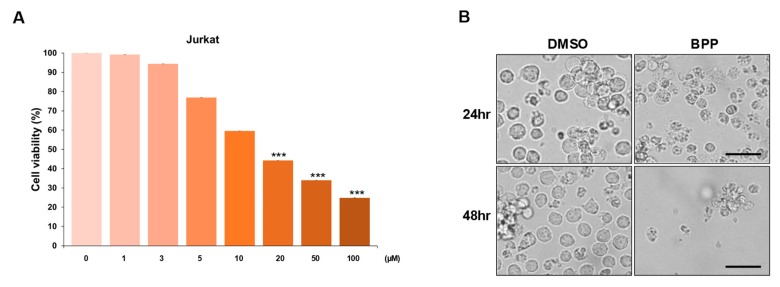
(**A**) BPP-induced antiproliferative effects on Jurkat cells. Cell viability was determined using an MTS assay 48 h post treatment. (**B**) The BPP (20 μM)-treated Jurkat cells showed morphological changes characteristic of apoptosis. Phase contrast microscopy images (400× magnification) showing morphological changes 24 and 48 h post treatment. Live cells are round in shape.

**Figure 5 molecules-24-02960-f005:**
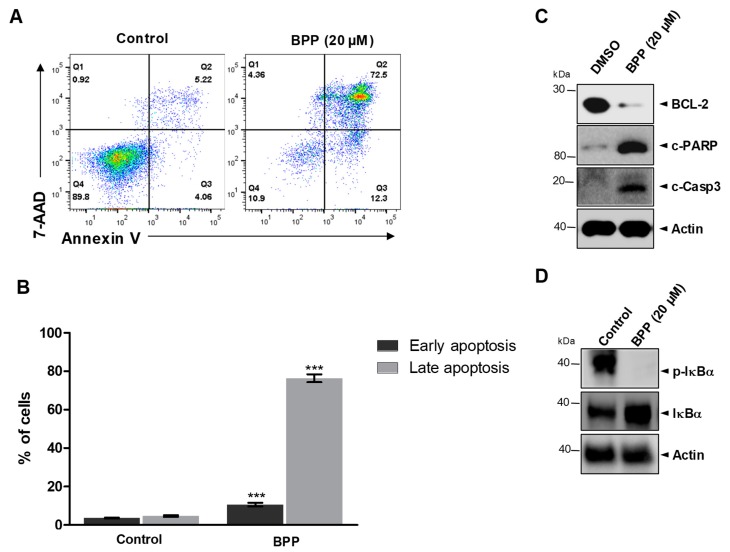
BPP-induced apoptosis in Jurkat cells. (**A**) Jurkat cells were compound-untreated or treated and apoptosis was evaluated 48 h thereafter using annexin V/7AAD double staining. Cell death was assessed by flow cytometry. (**B**) Early and late apoptotic cell rate quantification. (**C**) Whole cell lysates from BPP-treated cells were immunoblotted with antibodies specific for BCL-2, cleaved PARP, and cleaved caspase 3 proteins. (**D**) Whole cell lysates were used to determine the expression levels of I-kappa-B-alpha (IκBα) and p-IκBα after treating cells with BPP (20 μM).

**Figure 6 molecules-24-02960-f006:**
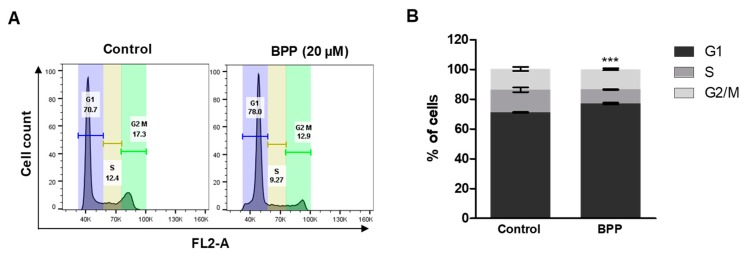
BPP-induced changes in cell cycle progression in Jurkat cells. (**A**) Cell cycle analysis of compound-treated cells. Cells were PI (propidium iodine)-stained for 30 min and analyzed by flow cytometry. (**B**) Percentage G1, S, G2/M phase cells were determined from triplicate experiments.

**Figure 7 molecules-24-02960-f007:**
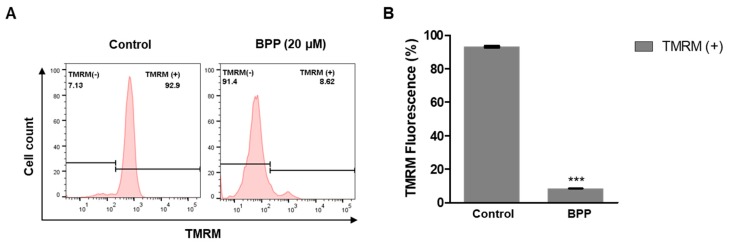
BPP-reduced Jurkat cell mitochondrial membrane potential. (**A**) Mitochondrial membrane potential of Jurkat cells loaded with TMRM (100 nM) and detected by flow cytometry. (**B**) Quantification of mitochondrial membrane potential.

**Figure 8 molecules-24-02960-f008:**
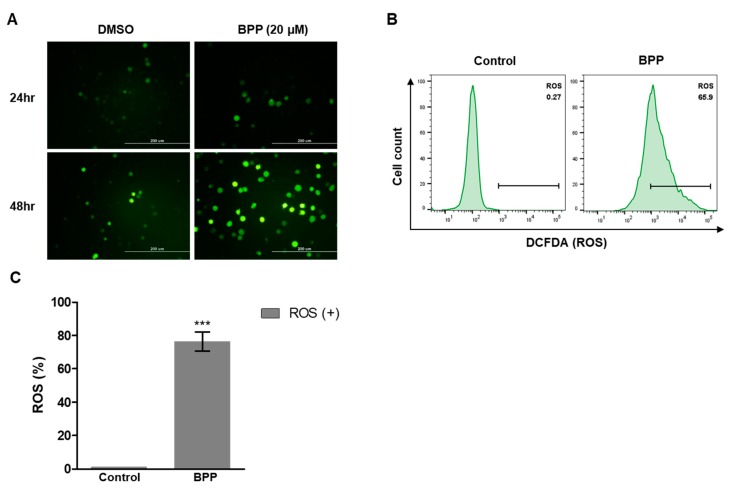
BPP-induced changes in intracellular ROS (reactive oxygen species) levels in Jurkat cells. (**A**) ROS detection by measurement of fluorescence intensity using fluorescence microscopy. (**B**) Jurkat cellular ROS levels were detected by flow cytometry with DCFDA (2′,7′-dichlorofluorescin diacetate) (1 μM). (**C**) Quantified ROS levels.

**Figure 9 molecules-24-02960-f009:**
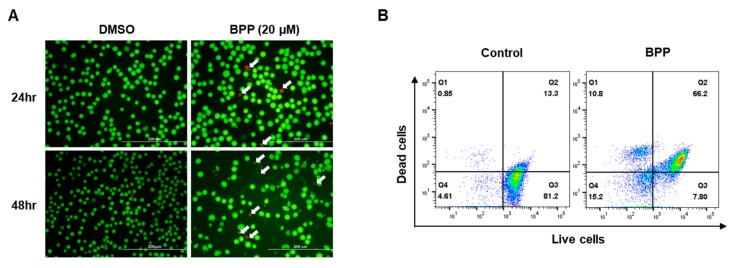
Measuring BPP-treated Jurkat cell viability. (**A**) Merged fluorescence images showing untreated and 48 h BPP-exposed Jurkat cells (green fluorescence, live cells; red fluorescence, dead cells). (**B**) Flow cytometry analysis of live and dead Jurkat cells with calcein and ethidium homodimer-1 staining.

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
