# Peer review of "Platyphylloside Isolated from Betula platyphylla is Antiproliferative and Induces Apoptosis in Colon Cancer and Leukemic Cells"

_molecules, 2019, doi:10.3390/molecules24162960_

Round 1

Reviewer 1 Report

Abstract

Line 22 “rings joined by a seven carbons chain”

Introduction

Line 32 “Cancer is the second leading cause of death in the United States” All authors are South Korean and I do not understand why they talking about the leading cause of death in the USA. What about South Korea? What is the rate of cancer diseases in South Korea? How does this study relate to the cancer situation in your country?

Lines 36 and 37 “

 Recently, CRC drug discovery 36 research efforts have focused on natural product compounds due to their limited side effects” the sentence needs to be sustained by a reference.

Line 39 “ Leukemia is categorized” please consider “It is categorized”

Lines 51 and 52 “Platyphylloside (BPP), s a well-known diarylheptanoid from the bark of Betula platyphylla (birch 51 tree), is widely distributed in Korea, Japan, China, Sahalin, and Siberia” replace by Platyphylloside (BPP), is a well-known diarylheptanoid from the bark of Betula platyphylla (birch tree), which is widely distributed in Korea, Japan, China, Sahalin, and Siberia

Line 53 “long known in traditional medicine in different parts of the world” long known in traditional medicine located in different parts of the world

Line 54 please consider “its most widespread uses have been in inflammatory diseases, including dermatitis, bronchitis…”

Results and discussion:

Lines 74 and 75 please consider “a large amount of BPP (Figure 1A) was isolated from”

Line 93 “Next, BPP-induced apoptosis was determined” please replace next by moreover,

Line 108 The authors compared the mechanism of BPP with those of corosolic acid. I did not understand the structural relationship between BPP and corosolic acid. One is a pentacyclic triterpenoic acid while the other is a diarylheptanoid glycoside.

In my opinion the author should consider compounds related to curcumin and its derivatives for the comparison to highlight whether or not their action on the cell cycle and the mechanism pathways are similar. Comparison with corosolic acid does not make sense for me.

There are few studies that can inspire your discussion. Please have a lock on these papers: Int. J. Mol. Sci. 2019, 20, 2454; doi:10.3390/ijms20102454 (Mechanism of Apoptosis Induced by Curcumin in Colorectal Cancer) and Asian Pac J Cancer Prev. 2014;15(1):93-100 (Curcumin induces caspase mediated apoptosis in JURKAT cells by disrupting the redox balance).

The authors should construct a strong discussion based on structure resemblance and former studies reported in the literature.

Author Response

1.Abstract Line 22 “rings joined by a seven carbons chain” We corrected the English in the manuscript according to your kindly comments. The corrected line now reads “rings joined by a seven-carbon chain” instead of “rings joined by a seven carbon chain 2.Introduction Line 32 “Cancer is the second leading cause of death in the United States” All authors are South Korean and I do not understand why they talking about the leading cause of death in the USA. What about South Korea? What is the rate of cancer diseases in South Korea? How does this study relate to the cancer situation in your country? The United States is a highly developed country with a large population; therefore, we wanted to emphasize how cancer seriously affects a developed country like the USA. Furthermore, in Korea, cancer accounted for one in four deaths, and more than 200,000 new cancer cases were diagnosed in 2014. In 2015, cancer was the ranked first in a list of top 10 leading causes of death in Korea, and a total of 214,701 new cases of cancer were diagnosed. Among these, colon and rectal cancer were ranked the third or fourth cancer leading to death in both men and women (8,301 deaths), followed by lung (17,399 deaths), liver (11,311 deaths), and stomach cancer (8.526 deaths). The above information was reported by: Jung, K.W.; Won, Y.J.; Kong, H.J.; Lee, E.S. Cancer statistics in Korea: incidence, mortality, survival, and prevalence in 2015. Cancer res. Treat. 2018, 50(2), 303. We added the following text to the introduction as follows. “In Korea, cancer headed the list of top 10 leading causes of death, and colorectal cancer (CRC) is the third most common cancer in men and women, with more than one million cases diagnosed each year worldwide [2]. 3.Lines 36 and 37 “ Recently, CRC drug discovery 36 research efforts have focused on natural product compounds due to their limited side effects” the sentence needs to be sustained by a reference. We have added this reference in the revised manuscript to support this statement. Wong, K.E.; Ngai, S.C.; Chan, K.G.; Lee, L.H.; Goh, B.H.; Chuah, L.H. Curcumin nanoformulations for colorectal cancer: A Review. Frontiers Pharmacol. 2019, 10, 152. 4.- Line 39 “ Leukemia is categorized” please consider “It is categorized” - Lines 51 and 52 “Platyphylloside (BPP), s a well-known diarylheptanoid from the bark of Betula platyphylla (birch 51 tree), is widely distributed in Korea, Japan, China, Sahalin, and Siberia” replace by Platyphylloside (BPP), is a well-known diarylheptanoid from the bark of Betula platyphylla (birch tree), which is widely distributed in Korea, Japan, China, Sahalin, and Siberia - Line 53 “long known in traditional medicine in different parts of the world” long known in traditional medicine located in different parts of the world - Line 54 please consider “its most widespread uses have been in inflammatory diseases, including dermatitis, bronchitis…” - Results and discussion: Lines 74 and 75 please consider “a large amount of BPP (Figure 1A) was isolated from” Line 93 “Next, BPP-induced apoptosis was determined” please replace next by moreover, We have now corrected our English language errors based on your kind comments and suggestions. - Line 44: “Leukemia” has been changed to “It” - Line 57 and 59: We corrected this sentence and have changed it to “A platyphylloside denoted as BPP (Betula platyphylla platyphylloside) is a well-known diarylheptanoid obtained from the bark of Betula platyphylla (birch tree), which is widely distributed in Korea, Japan, China, Sahalin island, and Siberia [8].” - Line 59: The phrase now reads as “healing properties of B. platyphylla bark have long been known in traditional medicine located in different parts of the world” - Line 60: “Its most widespread uses have been” has now been used. Thank you for your suggestion. - Line 82: We have changed “a large amount of BPP” instead of “large amounts of” - Line 153: “Next” has been changed to “Moreover” In addition, our manuscript underwent extensive English editing by English editing service during revision for checking grammar, spelling and some improvement of style. We appreciate for reviewer’s kindly comments. 5.Line 108 The authors compared the mechanism of BPP with those of corosolic acid. I did not understand the structural relationship between BPP and corosolic acid. One is a pentacyclic triterpenoic acid while the other is a diarylheptanoid glycoside. In my opinion the author should consider compounds related to curcumin and its derivatives for the comparison to highlight whether or not their action on the cell cycle and the mechanism pathways are similar. Comparison with corosolic acid does not make sense for me. There are few studies that can inspire your discussion. Please have a lock on these papers: Int. J. Mol. Sci. 2019, 20, 2454; doi:10.3390/ijms20102454 (Mechanism of Apoptosis Induced by Curcumin in Colorectal Cancer) and Asian Pac J Cancer Prev. 2014;15(1):93-100 (Curcumin induces caspase mediated apoptosis in JURKAT cells by disrupting the redox balance). The authors should construct a strong discussion based on structure resemblance and former studies reported in the literature. Thank you for your kind discussion. As requested by the reviewer, the cytotoxic results were compared with curcumin; both platyphylloside (BPP) and curcumin induced apoptosis targeting G0/G1 cell cycle arrest in colon cancer and leukemia cells (Lim et al., 2014). Furthermore, it was reported that the induction of apoptosis by curcumin on Jurkat cells takes place via the induction of intracellular ROS followed by the disruption of the mitochondrial membrane potential. These results are similar to those observed with BPP on the Jurkat cells and reported in our manuscript (Gopal et al., 2014). With regard to the chemical structure, BPP and curcumin are diarylheptanoids, which are composed of two aromatic rings joined by a seven-carbon chain. The only difference was the presence of a glycoside moiety instead of carbonyl group at C-5 and the absence of two methoxy groups in the aromatic rings and double bonds in the side chain. In terms of structure-activity relationships, it was reported that BPP, which has a keto-enol moiety and a hydroxyl group in the aromatic ring, showed more potent cytotoxic activity on cancer cells than the other types of diarylheptanoids (Choi et al., 2008). Furthermore, according to the in vitro results, compared with diarylheptanoids, suggested that the coexistence of the glucosyl group at C-5 and a carbonyl group at C-3 in these diarylheptanoids was important and confers significant activities and selectivity towards cancer cells (Novaković et al., 2014). In our previous study, BPP exhibited considerable cytotoxic potential on 60 cancer cell lines and selectivity towards colon cancer and leukemia cells. Therefore, in this study, we focused on evaluating BPP effects in colon cancer and leukemia and investigating the possible mechanism of action. * Additions - Description of the discussion in text: line 84-87 of page 2 - Description of the discussion in text: line 173-178 of page 4 - Description of the discussion in text: line 221 of page 6 - Added reference [26] in line 221 of page 6 Reference: Choi, S. E., Kim, K. H., Kwon, J. H., Kim, S. B., Kim, H. W., & Lee, M. W. (2008). Cytotoxic activities of diarylheptanoids from Alnus japonica. Archives of pharmacal research, 31(10), 1287. Novaković, M., Pešić, M., Trifunović, S., Vučković, I., Todorović, N., Podolski-Renić, A., ... & Milosavljević, S. (2014). Diarylheptanoids from the bark of black alder inhibit the growth of sensitive and multi-drug resistant non-small cell lung carcinoma cells. Phytochemistry, 97, 46-54. Lim, T. G., Lee, S. Y., Huang, Z., Chen, H., Jung, S. K., Bode, A. M., ... & Dong, Z. Curcumin suppresses proliferation of colon cancer cells by targeting CDK2. Cancer Prevention Research 2014, 7(4), 466-474. Gopal, P. K., Paul, M., & Paul, S. Curcumin induces caspase mediated apoptosis in JURKAT cells by disrupting the redox balance. Asian Pac J Cancer Prev 2014, 15(1), 93-100.

Reviewer 2 Report

The submitted manuscript describes the biological testing of Platyphylloside (BPP) in a range of assays against colon and blood cancer cell lines. The first part of the study discusses the effects on colon cells and effectively shows that at anything but reasonably high concentrations there is no effect. The graph on fig 2 is totally misleading as it attempts to show a significant difference when the graph compares the difference between 0 and 4% - which is not a meaningful change. 

The results for the blood cancers are more promising but again on Fig 4 it is clear that very large doses are required (100uM) to gain reasonable effect. The most interesting section was the study on apoptosis (seemingly at 20uM) which I believe should be further examined. The methods are appropriately described and the results appear valid in the context of the work done. 

However I feel that the small findings reported here to do justify publication at this time. I wider study on a range of analogues of this compound to determine an SAR, or else a wider range of cell lines to truly determine if blood cancers are a specific cancer target for this compound should be required prior to publication.

Additional Notes:

1. I am unsure why Platyphylloside has been abbreviated to BPP, this is confusing.

Author Response

The submitted manuscript describes the biological testing of Platyphylloside (BPP) in a range of assays against colon and blood cancer cell lines. The first part of the study discusses the effects on colon cells and effectively shows that at anything but reasonably high concentrations there is no effect. The graph on fig 2 is totally misleading as it attempts to show a significant difference when the graph compares the difference between 0 and 4% - which is not a meaningful change. We thank the reviewer for the thoughtful comments and suggestions. We agree that apoptosis with 10 μM BPP treatment may not be very strong. Therefore, we optimized the BPP concentration using the MTS assay. Our data showed that BPP conditions above 20 μM are most suitable for assessment (Figure 1B). Microscopic analysis also indicated that BPP treatment induced morphological changes compared to dimethyl sulfoxide (DMSO) treatment (Figure 1C). Moreover, 20 μM BPP increased the percentage of apoptotic RKO cells by more than 50 % (Figure 2A, 2B). Therefore, BPP effectively induced apoptosis in RKO cells. The results for the blood cancers are more promising but again on Fig 4 it is clear that very large doses are required (100uM) to gain reasonable effect. The most interesting section was the study on apoptosis (seemingly at 20uM) which I believe should be further examined. The methods are appropriately described and the results appear valid in the context of the work done. We appreciate thoughtful comments and suggestions. While revising our manuscript, we further investigated the antiproliferative effects of 20 µM BPP in colon cancer cells. In our experiments, we confirmed that the IC50 value was approximately 20 µM as per the results of the MTS assays in RKO and Jurkat cells. Hence, we investigated the mechanism of action, including apoptosis, cell cycling, ROS, and mitochondrial membrane potential, with 20 µM of BPP. Our results demonstrated that 20 µM BPP showed significant cytotoxic effects via apoptotic mechanisms, in both RKO and Jurkat cells. However I feel that the small findings reported here to do justify publication at this time. I wider study on a range of analogues of this compound to determine an SAR, or else a wider range of cell lines to truly determine if blood cancers are a specific cancer target for this compound should be required prior to publication. We sincerely appreciate the reviewer’s thoughtful comments and helpful suggestions. While revising our manuscript, we further investigated the possible mechanisms of the anti-proliferative actions of BPP and assessed cleaved caspase 3, I-kappa-B-alpha (IκBα), and p-IκBα in both RKO and Jurkat cells. We agree with the reviewer’s opinion regarding testing the cell-specific effects of BPP in colon and blood cells. In fact, we designed this study based on the reviewer’s comment. We have already tested 60 different kinds of cancer cell lines with BPP. Our previous results provided information that BPP could be a good anti-cancer agent, targeting leukemia or colon cancer cells. These results are well described in our previous reports (Cho et al., 2016). However, since our reports, no further studies about the mechanism action of apoptosis with BPP in colon or leukemia cells have been reported. Therefore, we performed the current study and emphasized the motivation of our current study in the introduction of manuscript. Cho, N.; Woo Kim, H.; Bum Kim, T.; Ransom, T.T.; Beutler, J.A.; Sung, S. Preparative purification of anti-proliferative diarylheptanoids from Betula platyphylla by high-speed counter-current chromatography. Molecules 2016, 21(6), pp. 700. * Additions -Modification of Abstract in text: line 19-22 of page 1 - Description of “the motivation of current study” in text: line 65-73 of page 2 In addition, according to reviewer’s comments, we employed a wider range of cell lines to demonstrate the anti-proliferative effects of BPP in leukemia and colon cancer cell lines and added the MTS results in the supplementary information. Additional Notes: 1. I am unsure why Platyphylloside has been abbreviated to BPP, this is confusing. In our previous study, we established a method for the simultaneous determination of platyphylloside for the quality control of Betula platyphylla bark using high performance liquid chromatography and a diode-array UV/V detector (HPLC-DAD) (Cho et al., 2014). Validation was successfully applied to determine the content of platyphylloside in three batches of Betula platyphylla bark extract. We proved that the platyphylloside is one of the major components of the Betula platyphylla bark. In our experiments, the average content (%) of platyphylloside in the three batches was 3.75%, 2.57%, and 2.84%. This indicates that platyphylloside is one of the major components of the Betula platyphylla bark. In addition, we suggested a simple and rapid isolation method using high-speed counter-current chromatography (HSCCC) (Cho et al., 2016). Hence, we wanted to emphasize that platyphylloside is the major compound and is obtained from Betula platyphylla bark. Lee et al. (2016) also used the term BPP, which is the abbreviated form of Betula platyphylla platyphylloside. Cho NK, Kim DH, Sung SH. Simultaneous Determination of Platyphylloside, Aceroside VIII and Betulin in Betula platyphylla bark by HPLC-DAD. Kor. J. Pharmacogn. 45(4) : 294∼299 (2014) Lee M, Sung SH. Platyphylloside Isolated From Betula platyphylla Inhibit Adipocyte Differentiation and Induce Lipolysis Via Regulating Adipokines Including PPARγ in 3T3-L1 Cells. Pharmacogn Mag. (2016) Dec;12(48) We appreciate this point as we failed to make this observation in our previous submission. We have added the full name in the introduction to reduce confusion and improve the clarity. * Additions - Description of the full name in text: line 57 of page 2

Reviewer 3 Report

Lee and Thuy at al. have shown induction of apoptosis in leukemia cells and minor increase in apoptosis of colon cancer cells after BPP treatment. Authors have also shown ROS and cell cycle arrest with BPP treatment. Authors need to interrogate the effect of BPP treatment systematically to establish its mechanism of action. I have several minor and major comments below:

1.      Figure 1B, 10μM BPP concentration showed stronger cytotoxicity in RKO than WiDr and HT-29. It shows cell line specific effect of BPP. It is important to show the cytotoxic effect of BPP in multiple cell lines to rule out any cell line specific effect.

2.      IC50 of BPP appear to be more than 10μM in vitro in colon cancer cell lines. What concentration can we achieve in vivo to show therapeutic efficacy of BPP without systemic toxicity.

3.      Figure 2B, 4% apoptosis with 10μM BPP treatment is clinically not significant. Need to show in vivo efficacy of BPP.

4.      Figure 2C, need to show band of cleaved caspase 3 with BPP treatment.

5.      Please explain the reason why authors have used 5X and 10X concentration of BPP in leukemia cells as compared to colon cancer cells.

6.      Figure 5C, please show the band of cleaved caspase 3 as well.

7.      Mechanistically it is unclear how BPP is inducing its anti-proliferative effect in colon and leukemia cancer cells.

8.      Authors need to connect cell cycle arrest, ROS and apoptosis induction by BPP treatment in a sequential way.

9.      BPP treatment is not very effective in colon cancer cells and effective at high concentration in leukemia cells. In vivo experiment is highly desired to show the therapeutic potential of BPP.

Author Response

Figure 1B, 10μM BPP concentration showed stronger cytotoxicity in RKO than WiDr and HT-29. It shows cell line specific effect of BPP. It is important to show the cytotoxic effect of BPP in multiple cell lines to rule out any cell line specific effect. We sincerely appreciate the reviewer’s thoughtful comments and helpful suggestions. We agree with the reviewer’s opinion with regard to testing whether BPP has a cell line-specific effect or not. To investigate if BPP demonstrates a cell-specific effect among colon cancer cells, we performed the MTS assay using multiple colon cancer cell lines. The anti-proliferative effect increased with BPP treatment in a dose dependent manner and according to the treatment duration (48 h). High doses of BPP (from 50 µM to 100 µM) induced anti-proliferation in colon cancer cell lines; but these were not significantly different. However, at the lower concentration of 20 µM, which is near the IC50, the MTS results varied among the colon cancer cells. As cell viability was significantly reduced in the RKO colon cancer cell line after BPP (20 µM) treatment, we performed further mechanism studies in RKO colon cells with 20 µM of BPP. * Additions -Modification of Abstract in text: line 19-23 of page 1 - Description of the results in text: line 91-96 of page 2 - Data: Supplement Fig 1. Fig. 2. 2. IC50 of BPP appear to be more than 10μM in vitro in colon cancer cell lines. What concentration can we achieve in vivo to show therapeutic efficacy of BPP without systemic toxicity. Our previous results with NCI 60 cancer cell lines proved that BPP showed potent cytotoxicity in colon cancer and leukemia cell lines, demonstrating specific effects on particular cell lines (Cho et al., 2016) Cho, N.; Woo Kim, H.; Bum Kim, T.; Ransom, T.T.; Beutler, J.A.; Sung, S. Preparative purification of anti-proliferative diarylheptanoids from Betula platyphylla by high-speed counter-current chromatography. Molecules 2016, 21(6), pp. 700. In addition, Lee et al., 2012, reported that BPP at concentrations from 50 to 100 µM significantly protected HT22 cells against neurotoxicity induced by glutamate insult; this provides evidence that BPP is a quite stable natural compound. In this report, platyphylloside (1 or 2 mg/kg body weight) significantly ameliorated scopolamine-induced amnesia in the passive avoidance test. In line with previous reports, our data could suggest an in vivo application in the future, although the exact concentration range should be investigated using in vivo experiments. Lee, K.Y.; Jeong, E.J.; Huh, J.; Cho, N.; Kim, T.B.; Jeon, B.J.; Kim, S.H.; Kim, HP.; Sung, S.H. Cognition-enhancing and neuroprotective activities of the standardized extract of Betula platyphylla bark and its major diarylheptanoids. Phytomedicine 2012, 19(14), pp. 1315-1320. 3. Figure 2B, 4% apoptosis with 10μM BPP treatment is clinically not significant. Need to show in vivo efficacy of BPP. We appreciate the careful comments with regard to our manuscript. As pointed out by the reviewer, we agree that the apoptosis experiment with 10 μM BPP treatment may not be a very strong treatment. Therefore, we performed a further apoptosis analysis with 20 μM BPP during revision duration and inserted new results and figures (Figure 1 to 5). Our data showed that BPP conditions above 20 μM are the most suitable (Figure 1B). Microscopic analysis also showed that BPP treatment induced morphological changes compared to dimethyl sulfoxide (DMSO) treatment (Figure 1C). Moreover, the percentage of apoptotic RKO cells was increased by more than 40 % under the 20 μM BPP treatment (Figure 2A, 2B). Our in vitro results suggest that BPP, a natural product, could be a potential multi-target therapeutic agent for leukemia and colon cancer; however, further in vivo experiments are necessary for drug development. 4. Figure 2C, need to show band of cleaved caspase 3 with BPP treatment. We appreciate the comment and have now shown the band of cleaved caspase 3 with BPP treatment as follows; 5. Please explain the reason why authors have used 5X and 10X concentration of BPP in leukemia cells as compared to colon cancer cells. While revising our manuscript, we further investigated the anti-proliferative of BPP at concentrations from 0 to 100 µM in both colon and leukemia cells. Accordingly, we have added new figures supporting the information. 6. Figure 5C, please show the band of cleaved caspase 3 as well. We appreciate the reviewer’s comment and have now shown the band of cleaved caspase 3 with BPP treatment as follows; 7. Mechanistically it is unclear how BPP is inducing its anti-proliferative effect in colon and leukemia cancer cells. We sincerely appreciate the reviewer’s comment. NF-kB is a critical transcription factor that regulates many genes associated with tumorigenesis. Thus, we investigated the relationship of this signaling with the molecular mechanism underlying BPP-induced cell apoptosis. The data indicated that the phosphorylation level of the inhibitory protein, IkBα, which binds to the NF-kB heterodimer (p50/p65), was reduced by BPP (Figure 2 D). This result suggests that BPP inhibited NF-kB signaling in RKO cells. 8. Authors need to connect cell cycle arrest, ROS and apoptosis induction by BPP treatment in a sequential way. Thank you for the thoughtful comment. Our results reported that BPP treatment rapidly and efficiently induced apoptosis in multiple colon and blood cancer cells. This process is dependent on cleaved caspase 3. Loss of checkpoint control, through the cell cycle, is involved in cancer progression. Cell cycle arrests participate in the anti-cancer process of many drugs and targeting the cell cycle is an important approach in cancer therapy (ref: Schwartz GK, Shah MA. Targeting the cell cycle: a new approach to cancer therapy. J Clin Oncol 2005; 23: 9408–9421.) Thus, we investigated the effects of BPP on the cell cycle, suggesting that cell cycle arrest plays an important role in the inhibitory effect of BPP in the colon and blood cancer cells. Many types of cancer cells exhibit increased levels of ROS. ROS production has proved to be associated with apoptosis and cell cycle arrest caused by anti-cancer agents. Accordingly, apoptosis and cell cycle arrests corresponded with the accumulation of ROS, suggesting its importance in the anti-cancer activity of BPP. In this study, our novel discovery demonstrated that BPP is a potent agent that induces cell cycle arrest and apoptosis in colon and blood cancer cells, through an NF-κB- and ROS-dependent pathway. Depending on kindly reviewer’s comment, we revised and added more discussion to explain the sequential correlation between cell cycle arrest, apoptosis, and ROS induced by BPP treatment as follows in the manuscript. * Additions - Description of the discussion in text: line 241-242 of page 7 - Modification of conclusion in text: line 361-370 of page 10 9. BPP treatment is not very effective in colon cancer cells and effective at high concentration in leukemia cells. In vivo experiment is highly desired to show the therapeutic potential of BPP We appreciate the point that we havn’t see. We agree that more in vivo experiments are required for anti-cancer drug discovery. Our current study have been designed to focus on investigating the anti-apoptotic mechanism study of BPP in vitro. We were able to isolate enough platyphylloside (BPP) from Betula platyphylla bark extract by the yield of 5-10%, since we developed efficient high speed counter current chromatography (HSCCC) method before as we describe in our manuscript. However, the amount of BPP might not be enough for more in vivo test. Instead, we improved our results with various colon and leukemia cell lines at various concentrations during revision period. We also showed more anti-proliferative mechanism including cleaved caspase3, IκBα, p-IκBα signaling and so on according to reviewer’s kindly comments. Combined with our previous reports, this reports could suggest that BPP could be a potential multi-target therapeutic agent for leukemia and colon cancer, and could be worthy of reference in future for further in vivo experiments for the anti-cancer drug development with BPP.

Round 2

Reviewer 2 Report

The revised manuscript address my original concerns and is significantly advanced on the first submission. The additional cellular data (whole cell and assays) and revised discussion of the results now make this revision suitable study for publication.

Author Response

We thank the reviewers for the thoughtful comments and suggestions

Reviewer 3 Report

Authors have satisfactorily addressed my concerns. 

Author Response

(The authors gave the same response as above.)
